# High Refractive Index GRIN Lens for IR Optics

**DOI:** 10.3390/ma16072566

**Published:** 2023-03-23

**Authors:** Yan Kang, Jin Wang, Yongkun Zhao, Xudong Zhao, Haizheng Tao, Yinsheng Xu

**Affiliations:** 1State Key Laboratory of Silicate Materials for Architectures, Wuhan University of Technology, Wuhan 430070, China; 2Research Center, Nanjing Wavelength Optoelectronic Technology Co., Ltd., Nanjing 211100, China

**Keywords:** chalcogenide glass, thermal diffusion, gradient refractive index, Raman spectrum, electron probe

## Abstract

Infrared gradient refractive index (GRIN) material lenses have attracted much attention due to their continuously varying refractive index as a function of spatial coordinates in the medium. Herein, a glass accumulation thermal diffusion method was used to fabricate a high refractive index GRIN lens. Six Ge_17.2_As_17.2_Se*_x_*Te_(65−*x*)_ (*x* = 10.5–16) glasses with good thermal stability and high refractive index (*n*_@10 μm_ > 3.1) were selected for thermal diffusion. The refractive index span (∆*n*) of 0.12 was achieved in this GRIN lens. After thermal diffusion, the lens still had good transmittance (45%) in the range of 8–12 μm. Thermal imaging confirmed that this lens can be molded into the designed shape. The refractive index profile was indirectly characterized by the structure and composition changes. The structure and composition variation became linear with the increase in temperature from 260 °C to 270 °C for 12 h, indicating that the refractive index changed linearly along the axis. The GRIN lens with a high refractive index could find applications in infrared optical systems and infrared lenses for thermal imaging.

## 1. Introduction

Optical lenses based on existing infrared materials cannot meet the lightweight design requirements (size, weight, and performance) for shipborne or portable imaging systems due to their large volume, heavy weight, and high cost. Accordingly, the development of infrared detection technology and the national defense demand for multi-spectral (shortwave and longwave infrared) simultaneous imaging has increased. The self-focusing GRIN lens, which can refract the light transmitted in the axial direction and gradually reduce the distribution of the refractive index along the radial direction, has been proposed for IR optics to reduce the weight and provide an additional degree of optical freedom for the design. An optical device known as a GRIN lens has refractive indices that continuously vary in the axial [1], radial [2], and spherical [3] dimensions. The refractive index of a GRIN lens is not a fixed constant [4], which is the main difference from conventional optics. The internal refractive index precisely controls the GRIN materials to produce optical qualities that are carefully tailored [5]. When combined with the refraction from the optical device surface, the internal GRIN features will provide optical designers with additional degrees of freedom. These features include additional thermal or chromatic aberration correction and lightweight design [5,6], reducing the number and weight of lenses by up to 85%. Consequently, GRIN materials have attracted more attention in many fields, such as national security, biology, and optical communications [7].

The GRIN lens has been observed in the eyes of humans and fish for optics in the visible spectrum. Element diffusion [8], ion exchange [9], electrospray printing [10], laser-induced vitrification [11], and controlled nanocrystal formation [2] have been used to fabricate small GRIN lenses for the infrared region. Infrared GRIN lenses have made significant progress in recent years. In 2014, Gibson et al. [12] from the US Naval Research Laboratory (NRL) made GRIN infrared lenses using the glass accumulation thermal diffusion method with 13 varieties of infrared glasses. An axial refractive index distribution was created. The shape and the change of refractive index of the GRIN lens can be efficiently controlled by using this method. In 2017, Richardson K. et al. [10] reported the direct printing of a mid-infrared transparent GRIN film by continuously depositing two chalcogenide glass components. Using electrospray printing technology, they deposited Ge_23_Sb_7_S_70_ and As_40_Se_60_ layers together, and achieved ∆*n* > 0.4. In 2019, GRIN materials were prepared by the heat-induced crystallization method of Yadav et al. [13]. After heat treatment for nucleation, the refractive index decreased from 2.8440 ± 0.0005 to 2.8309 ± 0.0005 at 4.515 μm and then increased to 2.8723 ± 0.0005 after growth heat treatment at 270 °C. However, this approach is more challenging because the starting material must be low-loss (high optical uniformity) to achieve refractive index modification beyond the starting material. In 2020, Zhang et al. [14] introduced a new method for preparing infrared GRIN lenses based on the spatially resolved crystallization method. A maximum refractive index difference of 0.032 ± 0.001 at 1.551 μm was measured between the substrate glass and the glass ceramic. However, crystallinity is challenging to control by temperature, making it difficult to achieve the desired refractive index span (∆*n*) and to prepare large-size GRIN samples. In 2022, Liu et al. [15] prepared an infrared axial GRIN lens by powder accumulation and spark plasma sintering (SPS). However, the SPS and extended grinding time require a significant investment of time and resources. Therefore, the glass accumulation thermal diffusion method reported by Gibson et al. [12] was adopted in this study. The shape of lens and refractive index of each glass layer can be adjusted by this method.

GRIN lens can be used for optimizing the volume and weight of the optical system by reducing the number of optical components. In addition to eliminating the advanced spherical aberration of the lens in the optical system, the optical path of the optical design can be effectively shortened by further increasing the refractive index of the infrared glass [16], thereby reducing the overall weight of the optical system. Therefore, improving the refractive index of the substrate glass is of great significance for GRIN lens research.

Chalcogenide glass is a good candidate for thermal imaging [17] and infrared optical lenses. Chalcogenide glasses have great potential for fabricating GRIN lenses by the glass accumulation thermal diffusion method due to their wide infrared transmission range (2–18 μm) and good thermal stability. In particular, the infrared transmittance range of Te-based glass can reach about 25 μm. Most research on GRIN infrared glasses is based on As_2_Se_3_ [18,19]. However, the problems of low refractive index and high glass dispersion have yet to be solved. For example, the NRL1 and NRL11–NRL23 glasses developed by Gibson et al. [20] have a maximum refractive index of only 2.7649 ± 0.0005 at 10 μm.

The linear refractive index of the infrared lens, which is closely related to the density and ion polarization [21] of the chalcogenide glass, must be improved to eliminate advanced spherical aberrations and expand the field of view and aperture angle of the optical device [22]. The Te-based chalcogenide glass has almost the highest refractive index among the chalcogenide glasses. For example, the refractive index of Ge_10_As_20_Te_70_ is 3.610 at 10 μm [23,24]. However, the glass formation decreases due to the strong metallicity [25] of Te atoms, and the tendency to crystallize increases. In our previous work, the glass stability was improved by introducing Se into the Ge–As–Te glass [16]. Therefore, the Ge_17.2_As_17.2_Se*_x_*Te_(65−*x*)_ (*x* = 10.5–16) glasses with good thermal stability and high refractive index (*n*_@10 μm_ > 3.1) were selected to fabricate the GRIN lens. As a result, a refractive index span ∆*n* > 0.12 was achieved. In the Ge–As–Se–Te glass, Se replaces Te given that Se and Te are in the same group in the periodic table, so both elements have the same coordination number of 2, maintaining the same mean coordination number (MCN = 2.516). Therefore, the replacement of Te by Se did not significantly change the glass transition temperature (*T*_g_). Thus, the thermal diffusion between these glasses can take place at the same temperature. When the *T*_g_ of the six glasses used for lens preparation are similar, the refractive index obtained varied in a gradient because the ionic polarization of Te is higher than that of Se.

## 2. Materials and Methods

### 2.1. Preparation of Substrate Glasses

The Ge_17.2_As_17.2_Se*_x_*Te_(65−*x*)_ (*x* = 10.5, 12, 13, 14, 14.5, and 16, in mol%) samples were prepared with an appropriate amount of high-purity (6N) Ge, As, Se, and Te. First, the raw materials were placed in a quartz ampoule (Φ = 10 mm), which was then sealed after being pumped to a vacuum of 10^−4^ Pa with a molecular pump. Then, the sealed quartz ampoule was heated to 850 °C in a rocking furnace for 24 h. After homogenization, the ampoule containing the melt was quenched in water to form a glass rod. Subsequently, the glass rod was then annealed in a muffle furnace near *T*_g_ for 2 h to release the internal stress. It is noteworthy that, in the process of glass quenching, it is necessary to pay attention to the separation of the glass and quartz tubes, and then the glass can be annealed after quenching. Finally, the glass rod was removed from the quartz ampoule, cut into thin slices (Φ = 10 mm, *h* = 1 mm) with a wire cutter machine, then 800#, 1200#, and 2000# sandpaper was used to smooth the surface of the glass and polished it to optical quality. The compositions and properties of the six glasses are listed in Table 1.

### 2.2. Diffusion under Pressure

The polished glass slices with different compositions were axially stacked in order of refractive index from high to low and first bonded together by heating to 260 °C. This step prevents uneven glass surfaces due to uneven pressure during diffusion. Next, the top and bottom of the glass were padded with graphite paper and transferred into the stainless-steel mold. Graphite paper has a high melting point (about 3850 °C), so it was chosen to prevent the glass from contaminating the mold when the temperature rises. Additionally, graphite paper can prevent mold scratches due to demolding failure. (the technological parameters of each sample are shown in Table 2). Then, the mold was transferred to the furnace chamber, and a vacuum pump was used to pump the pressure below 10 Pa. Finally, the mold was slowly heated to soft temperature, and a pressure of 7.5 kPa was applied to the upper die core to diffuse the glass layers and reduce the overall thickness. The glass was slowly removed after diffusion. Because an axial GRIN lens was prepared in this study, it is necessary to use a cutting machine to cut the glass along the axial direction and measure the cross−section of the glass. The hot-pressing diffusion process of the glasses and the GRIN lens molding process are shown in Figure 1.

### 2.3. Characterization

The densities of the samples were tested five times using the Archimedean drainage method, and then the average values were taken. The glass transition temperatures (*T*_g_) of these glasses were determined using differential scanning calorimetry (DSC; STA449F1, NETZSCH, Berlin, Germany). Under the protection of a N_2_ atmosphere, 20 mg powder samples were sealed in an aluminum crucible and tested in the temperature range of 20 °C–350 °C (heating rate: 20 K/min). The glass samples before and after thermal diffusion were measured with an X-ray diffractometer (XRD; D8 Discover, Bruker, Karlsruhe, Germany) in the range of 20°–70°, verifying the amorphous characteristics of the glass. The infrared transmission spectrum (7500–400 cm^−1^) of the glass slice was measured with a Fourier transform infrared spectrophotometer (FTIR; INVENIO S, Bruker, Ettlingen, Germany). The IR–variable angle spectroscopic ellipsometer (IR-Vase Mark II, J.A. Woollam Co., Ltd., Lincoln, NE, USA) which can achieve high-precision measurement with an error of ±0.001, was used to measure the refractive index (*n*) of the single-sided polished glass samples with a thickness of 1 mm. The Raman spectra of the hot-pressed glass samples were collected with a Raman spectrometer (LabRam HR Evolution, Horiba Jobin Yvon, Paris, France). A 532 nm laser was chosen as the excitation wavelength. The contents of Ge, As, Se, and Te were measured by line scanning of the GRIN samples with an electron probe microanalyzer (EPMA, JXA-8230, JEOL Ltd., Akishima, Tokyo, Japan).

## 3. Results and Discussion

### 3.1. Characterization of the Substrate Glasses

Figure 2a shows the DSC results of the Ge_17.2_As_17.2_Se*_x_*Te_(65−*x*)_ glass samples. The value of *T*_g_ was taken as the intersection of the baseline and the extension of the transition temperature tangent line. The results show that the *T*_g_ values of G1–G6 were close to each other, and ranged from 160 °C to 165 °C. This observation was attributed to the fact that Se and Te belong to the same group in the periodic table, and the replacement of Te by Se will not significantly change the *T*_g_. The slight difference in *T*_g_ ensures that the diffusion between the glass slides can occur at the same temperature. A simple measure of stability towards crystallization on reheating a glass to above *T*_g_ for shaping, like preform drawing, is given by the Hruby parameter [26]: [*T*_x_−*T*_g_] (where: *T*_x_ is the onset temperature of crystallization). As shown in Figure 2a, no crystallization peak appeared for all six glasses, indicating that these glasses can endure the prolonged thermal diffusion.

In Figure 2b, the transmittance of the glass was around 55%, and the transmittance range was 2–17 μm. For an optical sample with the thickness *h*, the transmittance of the light can be determined by the Lambert–Beer function [27,28,29] as follows:(1)$$T={\left(1-R\right)}^{2}e^{-\alpha h}$$
where *T* is the transmittance (%), *R* is the single surface reflection, *α* is the absorption coefficient, and *h* is the thickness of the optical material (cm).

The transmittance, including total internal reflectance, of a completely non-absorbing sample is as follows:(2)$$T=\frac{n}{n^{2}+1}$$
and the reflectance is given by:(3)$$R=\frac{{\left(n-1\right)}^{2}}{{\left(n+1\right)}^{2}}$$
where *R* is the total specular reflectance and *n* is the refractive index of the materials. Equations (1)–(3) present the relationship between the transmittance *T*, the reflectance *R*, and the refractive index *n*. Therefore, the reflectance increases with the refractive index, and the transmittance decreases accordingly. According to Equation (2), the maximum transmittance is only 60% when the refractive index reaches 3 or more. Meanwhile, the increase in thickness will lead to a decrease in transmittance.

Figure 3a shows that the refractive index of the Ge_17.2_As_17.2_Se*_x_*Te_(65−*x*)_ glasses gradually decreased with the increase in the Se content from 3.258 to 3.137 at 10 μm. The large difference in the refractive index between G1 and G6 gave a large gradient. The maximum refractive index difference between the adjacent two glasses was around 0.03 due to the lower ion polarization and relative atomic mass of the Se atom compared to the Te atom, resulting in a decrease in the refractive index. In Figure 3b, the refractive index (at 4 and 10 μm) gradually decreased with the increase in Se content.

As shown in Figure 4, the six substrate glasses underwent the Raman spectra test to investigate the changes in the glass structure. In the Ge–As–Se–Te glass, the Ge–Se bond will be preferentially formed when the Se is introduced into the Ge–As–Te glass [30]. The Se atom forms the mixed tetrahedral unit of [GeTe_4_Se_4−*x*_] by breaking the tetrahedral structure of [GeTe_4_], corresponding to the absorption vibration band at 136 cm^−1^. The vibration band at 116 cm^−1^ should be classified as the symmetric stretching vibration of the Ge–Te tetrahedron and the symmetric bending vibration of the As–Te trigonal cone [31,32].

Therefore, two prominent vibration bands, 116 and 136 cm^−1^, are related to the Ge–As–Se–Te glass. The vibration assignments of each structural unit corresponding to the GAST glass are summarized in Table 3.

### 3.2. Performance of GRIN Lens

The infrared transmittance of the sample polished after thermal diffusion is shown in Figure 5a. The long-wave infrared transmittance was still good (the decrease in transmittance was caused by the increase in thickness). However, the short-wave infrared transmittance was slightly decreased, mainly due to Rayleigh scattering caused by the introduction of impurities and interface defects between the glass slices [15]. In addition, the two wide diffusion halos in the XRD spectra confirmed that the GRIN samples were amorphous (Figure 5b).

The refractive profile was difficult to measure after diffusion. Therefore, we first performed the composition profile from top to bottom through line scan of energy dispersive spectroscopy (EDS). In Figure 6, the Se and Te changed linearly with the scanning distance rather than in a stepwise manner.

The structural evolution from the Raman spectra can represent the change in the glass composition, which can also reflect the refraction index profile. Figure 6b shows the Raman intensity distribution of the samples along the scanning direction through the 3D grid Raman spectra [33]. The variation of the intensity peak at 116 cm^−1^ can reflect the variation trend of the Te in the glass. The peak intensity at 116 cm^−1^ showed an approximately linear decrease from top to bottom. The variation trend of the peak can reflect the change of the glass composition along the axial direction, which also represents the change of refractive index along the axial direction.

The composition of the GRIN lens was then checked by EPMA scanning [34]. In the inset shown in Figure 7a, we can clearly distinguish the interface between each layer for the samples thermally diffused within a short period. The concentration gradually rose along the axis, indicating that the refractive index of the sample discontinuously increased. However, the good resolution of EPMA suggested that the profile of the refractive index can be represented by the Se contents. After diffusing for a long period (12 h), the interface between the glass layers was no longer visible, and the elements were linearly distributed along the axis, as shown in Figure 7b. The Se distribution gradually presented a linear distribution with the increase in diffusion temperature.

Figure 7c shows the EPMA test of the Ge, As, and Se concentrations with distance and the relationship between Raman peak intensity (116 cm^−1^) and distance. The comparison results showed that the EDS, Raman, and EPMA data were in good agreement. These three characterization methods reflected the axial variation of the refractive index of the GRIN lens well.

We combined the composition and refractive index of the six substrate glasses to determine the relationship between the Se content and the refractive index (10 μm) to confirm that the method mentioned above can characterize the distribution of the GRIN lens’ cross-section refractive index:(4)$$n_{10}=3.13046+\frac{0.14092}{e^{\left(C-12.84151\right)/1.03921}}$$
where *n*_10_ is the refractive index of the glass at 10 μm and *C* is the concentration of Se.

Then, the Se content obtained by the EPMA test was substituted into Equation (4), and the refractive index of each position in the cross-section of the GRIN lens was obtained by backward extrapolation. In Figure 7d, the refractive index variation of the cross-section of the GRIN lens showed a good correlation with that of the substrate glass within the error range. Therefore, the refraction distribution curve of the cross-section of the GRIN lens tended to be linear after thermal diffusion. However, the refractive index was not completely linear in this work. Therefore, future work will emphasized the optimization of composition, refractive index, and glass thickness.

We pressed a GRIN sample into a spherical lens and built a thermal-imaging set-up to observe the imaging of the GRIN lens [2] (Figure 8a). The set-up consisted mainly of a hot plate that illuminated a grid (glowing in the longwave infrared region). Then, the grid was imaged through the sample with a thermal imaging camera. It turns out that when the axial GRIN lens was used (Figure 8b), the grid took on its original shape. However, the grid was significantly deformed using a spherical GRIN lens, as shown in Figure 8c.

Abbe number is usually used to represent the dispersion of infrared materials [5,35]:(5)$$\nu =\frac{n_{\mathrm{center}}-1}{n_{\mathrm{short}}-n_{\mathrm{long}}}$$
(6)$$\nu_{\mathrm{GRIN}}=\frac{\Delta n_{\mathrm{center}}}{\Delta n_{\mathrm{short}}-\Delta n_{\mathrm{long}}}$$
where *n*_short_, *n*_center_, and *n*_long_ are the refractive indices at the short, long, and center wavelengths, and ∆*n* is the refractive index difference of a specific band. In this paper, the relationship between refractive index and Abbe number of the six samples in three wavebands, SWIR (2–3 μm), MWIR (3–5 μm), and LWIR (8–12 μm), was studied. Figure 9a shows the Abbe number and refractive index trends in the three bands. Similar trends were observed in the three bands. More importantly, at 10 μm, the Abbe number was more than 200, with low dispersion. Therefore, the dispersion of the whole optical system can be controlled effectively by the reasonable collocation of this series of glasses.

The dispersion can be expressed as [36]:(7)$$p=\frac{n_{\mathrm{center}}-n_{\mathrm{long}}}{n_{\mathrm{short}}-n_{\mathrm{long}}}$$
and
(8)$$p_{\mathrm{GRIN}}=\frac{\Delta n_{\mathrm{center}}-\Delta n_{\mathrm{long}}}{\Delta n_{\mathrm{short}}-\Delta n_{\mathrm{long}}}$$

We have mapped the P–V diagrams of common infrared glasses [37,38,39] and the GRIN materials prepared in this study in SWIR (2–3 μm), MWIR (3–5 μm), and LWIR (8–12 μm) bands, which are shown in Figure 9b and Table 4. We can see that the Abbe numbers of the six substrate glasses were positive at all three bands, while the GRIN sample was negative at all three bands. GRIN materials, along with diffractive optics, are one of the only means available to achieve a negative Abbe number. This is useful from the perspective of aberration, as such a GRIN profile will help to correct the uncorrected spherical aberration inherent in homogeneous elements [5]. Compared with common infrared materials, the dispersion of GRIN materials prepared in this study was very different from that of conventional infrared materials. Therefore, this study has effectively enriched the diversity of infrared materials.

## 4. Conclusions

In summary, a series of Ge_17.2_As_17.2_Se*_x_*Te_(65−*x*)_ glasses with a substantially unchanged *T*_g_ was prepared by gradually replacing Te with Se in the Ge–As–Te glass. The series glasses have a wide infrared transmission range (2–18 μm), good thermal stability, and high refractive index (*n*_@10 μm_ > 3.1). The Se diffused along the axial direction by stacking six glass pieces of continuously varying compositions together for diffusion. The refractive index distribution of the glass changed from step type to linear distribution. The refractive index span (∆*n*) of the GRIN lens was achieved at 0.12. The refractive index variation of the GRIN lens cross-section was compared with that of six substrate glasses. The refractive index of the GRIN lens was proven to be linearly distributed along the axis after diffusion.

Although the refractive index of the designed composition was not completely linear, the concentration of elements was linearly distributed over a prolonged diffusion. Accordingly, the refractive index distribution curve was considered to be linear. Furthermore, the linear distribution of the refractive index was achieved through the multi-layer stack.

## Figures and Tables

**Figure 1 materials-16-02566-f001:**
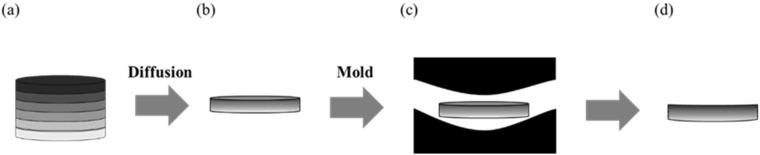
Schematic of the glass accumulation thermal diffusion method and molding. (**a**) Glass stack from the high refractive index (**top**) to low refractive index (**bottom**); (**b**) glass stack after thermal diffusion. The gradient of color indicates the distribution of refractive index. (**c**) Molding of GRIN lens into a spherical GRIN lens; (**d**) hot-pressed glass.

**Figure 2 materials-16-02566-f002:**
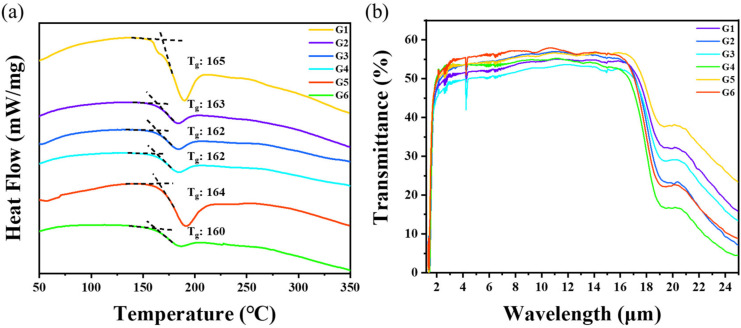
(**a**) DSC curves of the substrate glass at 10 K/min heating rate. (**b**) Infrared transmission spectra of the substrate bulk glass (1 mm thickness) at 1–25 μm.

**Figure 3 materials-16-02566-f003:**
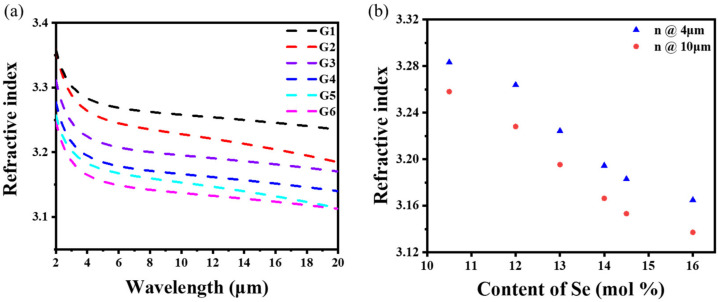
(**a**) Dependence of the refractive index of the Ge_17.2_As_17.2_Se*_x_*Te_(65−*x*)_ glasses on the wavelength; (**b**) the relationship between the refractive index and Se content.

**Figure 4 materials-16-02566-f004:**
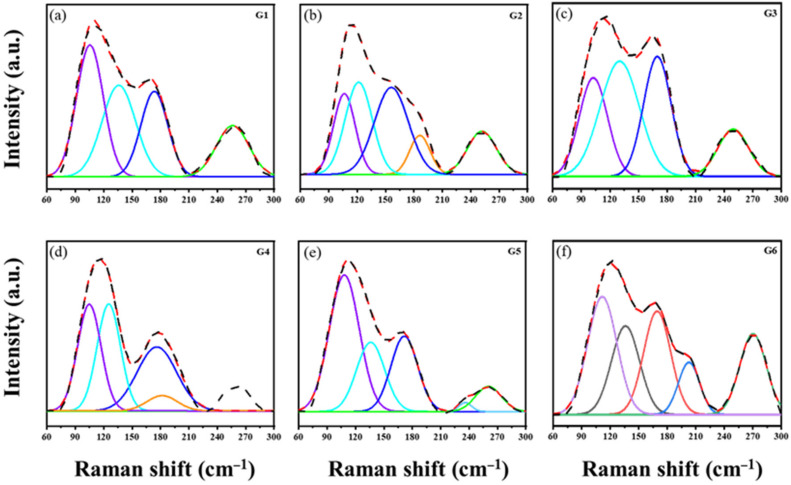
Raman spectra of (**a**) Ge_17.2_As_17.2_Se_10.5_Te_55.1_; (**b**) Ge_17.2_As_17.2_Se_12_Te_53.6_; (**c**) Ge_17.2_As_17.2_Se_13_Te_52.6_; (**d**) Ge_17.2_As_17.2_Se_14_Te_51.6_; (**e**) Ge_17.2_As_17.2_Se_14.5_Te_51.1_ and (**f**) Ge_17.2_As_17.2_Se_16_Te_49.6_.

**Figure 5 materials-16-02566-f005:**
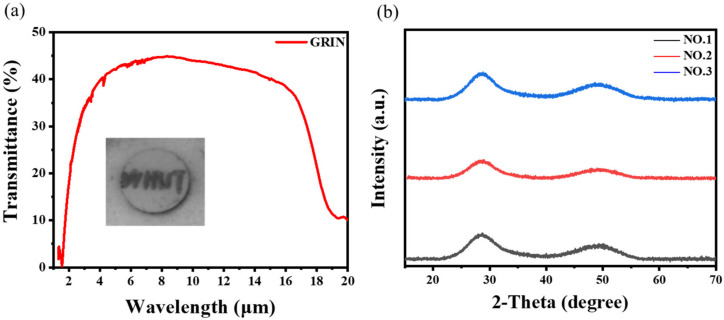
(**a**) Infrared transmittance of the samples after hot pressing diffusion polishing. The inset is an infrared photograph of the GRIN lens, thickness: 2.82 mm; (**b**) XRD spectra of the GRIN lenses.

**Figure 6 materials-16-02566-f006:**
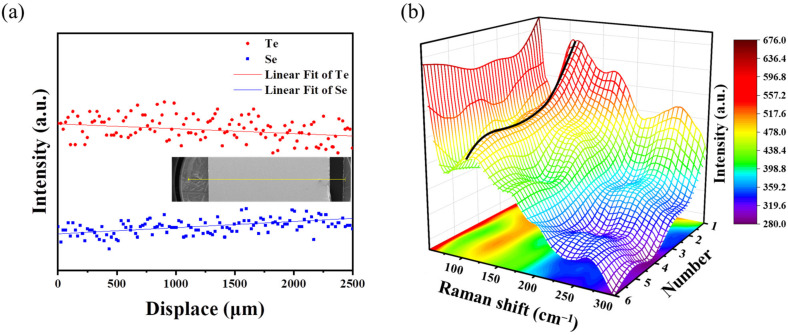
(**a**) EDS line scanning results of the GRIN lenses. The inset is the line scan direction; (**b**) Raman line scanning spectra of the GRIN lenses; numbers 1–6 in the figure represent the positions of the GRIN lens from top to bottom, and the color scale represents the intensity of the Raman scatting.

**Figure 7 materials-16-02566-f007:**
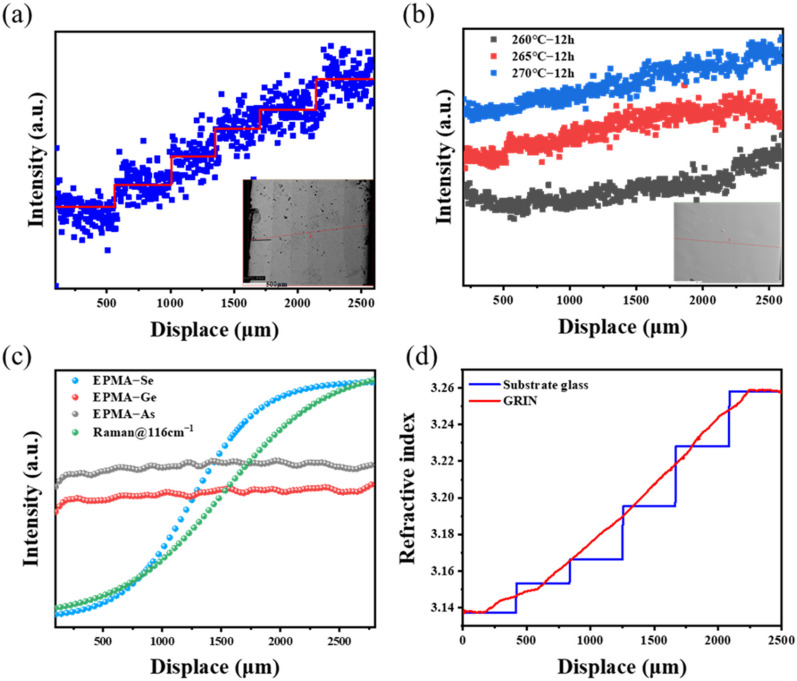
EPMA line scanning results of the two GRIN samples. (**a**) The element distribution of 20 min thermal diffusion (blue dot) and the corresponding refractive index distribution (red line); (**b**) thermal diffusion of 12 h, and it becomes linear with the temperature increase; (**c**) EPMA test of the Ge, As, and Se concentrations with distance and the relationship between Raman peak intensity (116 cm^−1^) and distance; (**d**) refractive index changes of the cross-section of the GRIN lens (270 °C–12 h) and the refractive index change of the substrate glass.

**Figure 8 materials-16-02566-f008:**
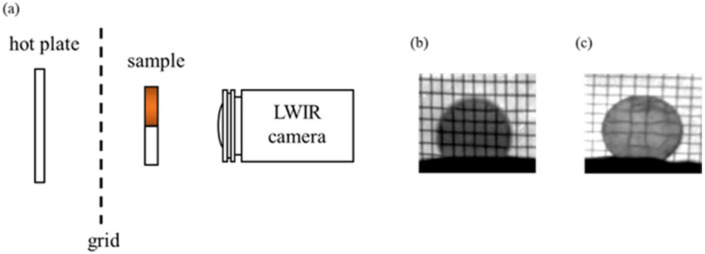
(**a**) Thermal imaging set-up for GRIN lens; (**b**) grid image of the axial GRIN lens; (**c**) under the same conditions, the grid of spherical GRIN lens was significantly deformed.

**Figure 9 materials-16-02566-f009:**
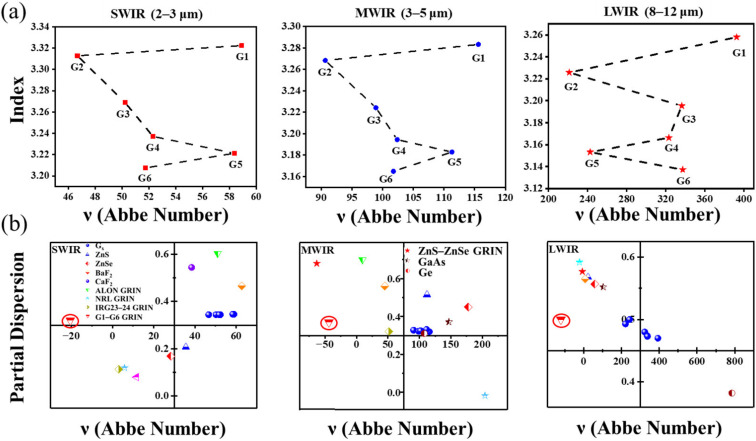
(**a**) Abbe number and refractive index trends for each band; (**b**) “P–V” diagrams of common infrared materials and GRIN material.

**Table 1 materials-16-02566-t001:** Glass composition, density (*ρ*), refractive index (*n*), Abbe number (*v*), glass transition temperature (*T*_g_), and transmittance (*T*%) of the GRIN lens at 10 μm.

Glass	Composition	*ρ* (±0.0001 g/cm^3^)	*n*_@10 μm_ (±0.001)	*v* _@10 μm_	*T*_g_ (°C)	*T* (±0.1, %)
G1	Ge_17.2_As_17.2_Se_10.5_Te_55.1_	5.3009	3.258	272	165	54.4
G2	Ge_17.2_As_17.2_Se_12_Te_53.6_	5.2656	3.228	153	163	56.5
G3	Ge_17.2_As_17.2_Se_13_Te_52.6_	5.2469	3.195	231	162	52.8
G4	Ge_17.2_As_17.2_Se_14_Te_51.6_	5.2195	3.166	221	162	54.8
G5	Ge_17.2_As_17.2_Se_14.5_Te_51.1_	5.2190	3.153	166	164	56.0
G6	Ge_17.2_As_17.2_Se_16_Te_49.6_	5.2126	3.137	230	160	57.3

**Table 2 materials-16-02566-t002:** Sample hot pressing process parameters.

Number	Temperature (°C)	Pressure (kPa)	Time (h)
1	260	7.5	12
2	265	7.5	12
3	270	7.5	12

**Table 3 materials-16-02566-t003:** Vibration assignments of the Raman peaks of the GAST glasses.

Raman Shift (cm^−1^)	Assignments of Vibrations
80–100	Te_3_ triangular cone antisymmetric stretching vibration
	AsTe_3_ triangular cone antisymmetric bending vibration
	Bending vibration of the GeTe_4_ tetrahedron
116	Symmetrical tensile vibration of the GeTe_4_ tetrahedron
	AsTe_3_ trigonal cone symmetric bending vibration
136	[GeTe_4_Se_4−*x*_] stretching vibration of a mixed tetrahedron
166	Antisymmetric bending vibration of the AsTe_3_ trigonal cone
167, 222, 235, 374	As–As bond vibration
183, 250	Ge–Ge bond vibration

**Table 4 materials-16-02566-t004:** Abbe number (V) and dispersion (P) of common infrared and GRIN materials.

Samples	SWIR		MWIR		LWIR	
	*V*	*P*	*V*	*P*	*V*	*P*
G1	58.91	0.346	115.61	0.320	392.53	0.470
G2	46.66	0.344	90.66	0.328	221.10	0.493
G3	50.21	0.344	98.90	0.322	336.35	0.474
G4	52.30	0.344	102.39	0.324	323.09	0.480
G5	58.36	0.346	111.29	0.332	242.55	0.500
G6	51.73	0.344	101.77	0.325	337.32	0.473
ZnS	35.60	0.207	112.05	0.515	22.87	0.568
ZnSe	28.22	0.168	177.70	0.450	57.84	0.557
G1−G6 GRIN	−20.50	0.321	−43.81	0.370	−120.9	0.500
Ge			107.51	0.311	784.54	0.382
GaAs			146.77	0.371	103.13	0.552
BaF_2_	62.72	0.466	45.04	0.560	7.14	0.565
CaF_2_	38.42	0.544				
ALON GRIN	51.00	0.603	9.37	0.700		
NRL GRIN	5.84	0.119	204.14	−0.017	−22.68	0.592
IRG23−24 GRIN	2.99	0.113	51.59	0.321		
ZnS−ZnSe GRIN	11.55	0.080	−63.13	0.680	−7.31	0.577

## Data Availability

The raw data required to reproduce these results cannot be shared at this time as the data also form part of an ongoing study.

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
