# Peer review of "High Refractive Index GRIN Lens for IR Optics"

_materials, 2023, doi:10.3390/ma16072566_

Round 1

Reviewer 2 Report

All tips, comments and questions to the authors of the paper were included in the comments in the pdf file sent to this review (24 comments in total).

Reviewer 3 Report

The authors report on the fabrication of GRIN lenses with high refractive index using the GeAsSeTe system with various compositions. Using thermal diffusion as part of the fabrication process, the lenses show a gradual variation in the refractive index.

In general, the manuscript is well organized, although it is recommended to look for typos and clarify some details that are not completely clear (see the suggestions). The authors show that their fabrication method is promising as a means to obtain GRIN lenses with the proposed materials operating in the spectral range from 8 to 12 microns. Some suggestions to improve the text, and some issues that must be addressed include the following:

1. Line 61. The statement "However, crystallinity is challenging to control by temperature, making it difficult to achieve the desired effect" is not fully clear. What would the desired effect be?

2.  Line 64. What does SPS stand for? Please define acronyms before using them.

3. In Table 1, please define n, v and Tg in the table header. Also, how were these parameters determined? 

4.  Line 117: "....together by heating to a certain temperature." What is this temperature? This must be included in the proper section as it is relevant to the fabrication process.

5. Line 119: "Then, the top and bottom of the glass were padded with graphite paper and transferred into the mold. Graphite paper can prevent mold contamination and damage due to release failure." What mold? This was not mentioned previously. Please describe the complete fabrication procedure. Also, regarding the graphite paper statement, is this a known fact? If so, proper references are required. Also, what is release failure? Please explain the meaning of this term (demanding perhaps?).

6. Line 157: "In chalcogenide glass, when ∆T > 100 °C, thermal stability is considered good. Hence, all the six substrate glasses have good thermal stability, long time thermal diffusion will not lead to crystallization of glass." How is this increase in temperature defined? Is this a general rule for this type of glass? If so, proper references must be cited. Note that the heat flow axis in the plot (Fig. 2) does not have a scale. Regarding the statement of long time thermal diffusion not leading to crystallization, how can this be infered from the results included in this section? Please provide the rationale of this and/or include proper references.

7. Line 164: "Infrared optical materials generally have strong absorption, resulting in low 164 transmittance." The transmittance of the fabricated glasses is around 55%. What happens to the rest of the optical energy? Is the light being absorbed? If so, is there any heating? What is the value of the scattering of the samples? These are very important aspects that must be discussed in this section of the manuscript. Also, what is the actual value of the transmittance for this materials? If a specific value is not included, it is hard to judge the improvements offered by the proposed glasses.

8. Regarding Raman characterization, please include the Raman spectra in the proper figure. Although the bands are indentified in Table 3, it is more illustrative for the reader to include the spectra.

9. Line 189: "The long-wave infrared transmittance is still good (the decrease in transmittance is caused by the increase in thickness). However, the short-wave infrared transmittance is slightly decreased, mainly due to Rayleigh scattering." Why is this considered “good”? With respect to what? Also, if the transmission decreases as the thickness of the sample increases, where do the losses come from? Absorption? Scattering? Both? These are important features that must be discussed and preferably quantified.

10. Line 200: "In Fig. 5, the Se and Te elements linearly change with the scanning 200 distance rather than change by step." This statement is not clear; what does this mean? The plot in FIg. 5(a) looks to remain constant rather than changing linearly. Also, in Fig. 5(b), what does the color scale represent?

11. Line 210: "The peak intensity at 116 cmshowed an evi- 210 dent linear decrease from top to bottom." This is not so evident from the plot. Perhaps it will be better to trace a line to remark this in the plot.

12. Line 219: "After diffusing for a long period (12 h), the interface 219 between the glass layers is no longer visible, and the elements are linearly distributed 220along the axis, as shown in Fig. 6(b)." Why not including as an inset the resulting sample as done in Fig. 6(a)? This will help to clarify this result.

13. Line 239: "Therefore, the refraction distribution curve of the cross-section of the GRIN lens is linear after thermal diffusion." Fig, 6(d) shows a continuos variation of the refractive index as a function of the position (or gradual, as desired in a GRIN lens) . The variation is not linear, it is in fact defined by eqn. 1. This figure shows that after thermal diffusion,  the refractive index indeed varies gradually, as also demonstrated in Fig. 6(b). I believe it woud be more interesting to compare the curves shown in 6(c) (blue and green) with the model given by eqn. (1). These points can be fitted to the equation and the fitting parameters will give information about the sample composition.

14. There is a typo in Fig. 6(a).

15. Line 298: "Combining the advantage of high refractive index while maintaining the same imaging effect, the lens size is significantly reduced. Moreover, the weight and size of the optical system are reduced." This is not demonstrated in the manuscript; strictly speaking, it is not supported by the results. To avoid any potential misleading, I would recommend rephrasing this statement or removing it from the conclusions.

Once the above issues are addressed, the manuscript can be considered for publication.
